# Guided Local Feature Matching with Transformer

**Siliang Du** [1,†], **Yilin Xiao** [1,†] , **Jingwei Huang** [1], **Mingwei Sun** [2] **and Mingzhong Liu** [1,*]

1   Huawei Technologies Co., Ltd., Wuhan 430074, China; dusi@whu.edu.cn (S.D.); xiaoyilin@whu.edu.cn (Y.X.); jingweih@stanford.edu (J.H.)
2   School of Remote Sensing and Information Engineering, Wuhan University, Wuhan 430072, China; mingweis@whu.edu.cn
*   Correspondence: mingzhongliu@foxmail.com
†   These authors contributed equally to this work.

**Abstract:** GLFNet is proposed to be utilized for the detection and matching of local features among remote-sensing images, with existing sparse feature points being leveraged as guided points. Local feature matching is a crucial step in remote-sensing applications and 3D reconstruction. However, existing methods that detect feature points in image pairs and match them separately may fail to establish correct matches among images with significant differences in lighting or perspectives. To address this issue, the problem is reformulated as the extraction of corresponding features in the target image, given guided points from the source image as explicit guidance. The approach is designed to encourage the sharing of landmarks by searching for regions in the target image with features similar to the guided points in the source image. For this purpose, GLFNet is developed as a feature extraction and search network. The main challenge lies in efficiently searching for accurate matches, considering the massive number of guided points. To tackle this problem, the search network is divided into a coarse-level match network-based guided point transformer that narrows the search space and a fine-level regression network that produces accurate matches. The experimental results on challenging datasets demonstrate that the proposed method provides robust matching and benefits various applications, including remote-sensing image registration, optical flow estimation, visual localization, and reconstruction registration. Overall, a promising solution is offered by this approach to the problem of local feature matching in remote-sensing applications.

**Keywords:** feature matching; guided points; guided point transformer; remote-sensing image registration; reconstruction registration

## 1. Introduction

Pairwise image feature matching aims to identify and correspond the same or similar content from two or more images at the pixel level. It is a fundamental problem in remote sensing, with various applications in remote-sensing image registration [1–4], change detection [5,6], and 3D reconstruction [7]. The most popular solution for image matching is to separately detect key points for the source and target images as feature points and establish matches between them, followed by traditional algorithms [8–12] and deep-learning-based approaches [13–16]. Some applications further select a subset of matched points sharing the same landmarks in space so that a global registration model can be derived. However, it is observed that image matching often encounters challenges at this stage, particularly when dealing with images exhibiting substantial differences in lighting or perspectives. Such discrepancies can lead to tracking failures in camera localization and incomplete results in 3D reconstruction, as separate landmarks detected from different images may not correspond to the same points in 3D space.

Some works address this challenge by designing detectors to produce more feature points [9,10,17,18], which increases the opportunity for sharing landmarks in both images.

However, it also raises the chance of producing false-positive matches and requires additional matching networks [11,12,15] to improve matching accuracy. In scenarios such as 3D reconstruction and visual localization, incremental matching based on existing models is often required. Leveraging the existing feature points can significantly improve efficiency and accuracy in such cases. As shown in Figure 1, the problem is reformulated by additionally considering guided points from the source image as guidance to regress corresponding points in the target image. It is argued that such a formulation encourages corresponding points from the image pair to share landmarks.

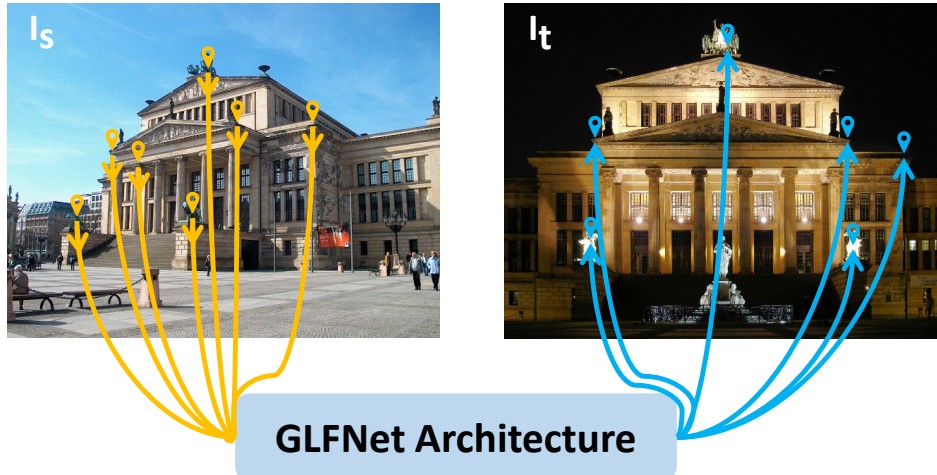

**Figure 1.** GLFNet is introduced to facilitate local feature matching between images, with guided points as the guidance. Diverging from the conventional approach of independently extracting and matching feature points from both images, the problem is reformulated by utilizing the guided points in the source image to guide the regression of exact coordinates for corresponding points in the target image. By adopting this novel perspective, the sharing of landmarks among matching pairs is fostered, thereby promoting more robust and accurate image matching.

For this reformulated problem, there are existing methods that can potentially address it. However, they suffer from certain drawbacks. One straightforward approach involves regressing the corresponding location in the target image for each feature point from the source image using [16]. However, this method involves iterative regression over the entire image, leading to increased runtime linearly with the number of guided points, which becomes time-consuming. Another brute-force solution exhaustively checks pixel-wise correspondences from a 4D correlation tensor [14]. Despite the fusion of features from two different resolutions in the 4D correlation tensor, the receptive field of 4D convolution remains limited to each match's neighborhood area. It is essential to highlight that both of these methods lack the ability to embed information from guided points in the feature representation process. To overcome these challenges, GLFNet is proposed to search for corresponding feature points in a coarse-to-fine manner based on the guided point transformer.

In detail, GLFNet consists of a feature extraction network and a search network. Each image is passed through the feature extraction network to produce coarse and fine feature maps. Further, the search network is split into a coarse-level match network and a fine-level regression network. The coarse-level match network embeds coarse image features to coarse patch features using the guided point transformer. Then, it matches the patches and guided points by solving an optimal transport problem based on the guided point transformer. Once the coarse matches are established, the fine-level coordinate regression network is utilized to regress the 2D coordinate for all guided points in its matched local patch based on fine feature maps. While [16] iteratively regresses the coordinate for each feature point in the entire image, the local search space supports the regression network to execute only once for all points but with higher accuracy, leading to acceleration by

173 times. As a result, the coarse-to-fine search network efficiently estimates accurate coordinates in the target image corresponding to guided ones in the source image.

The proposed method benefits various applications by introducing guided points. It outperforms the state of the art for remote-sensing image registration, optical flow estimation, and visual localization. By using GLFNet as the image-matching module, the standard pipeline of 3D reconstruction successfully incorporates more images into the camera graph. Based on guided points, it is found that GLFNet significantly improves alternative methods on an image-matching benchmark [19] in terms of accuracy.

Overall, the core contributions of the paper are:

- The image-matching problem is reformulated by considering guided points as input.
- GLFNet is designed with a coarse-to-fine search network to efficiently and accurately detect and match corresponding points. Additionally, the guided point transformer is proposed to incorporate the guided points information during feature representation.
- The GLFNet significantly improves the standard image-matching task and benefits various applications.

## 2. Related Work

### 2.1. Detector-Based Local Feature Matching

The detector is the key part of the detector-based local feature-matching method. Before the arrival of the deep learning era, SIFT [8] was the most successful hand-crafted local features detector, which is applied to a wide range of computer vision works. ORB [20] combines the advantages of FAST [21] and BRIEF [22] to efficiently complete feature extraction and feature description. Deep-learning-based methods can significantly improve performance in difficult environments. D2Net [9] is an approach where a single convolutional neural network plays a dual role: it is simultaneously a dense feature descriptor and a feature detector. ASLFeat [17] takes advantage of the inherent feature hierarchy to restore spatial resolution and low-level details for accurate keypoint localization. SuperPoint [10] proposes a self-supervised training method through homographic adaptation, which is currently the most widely used. GMS [11] encapsulates motion smoothness as the statistical likelihood of a certain number of matches in a region and enables translation of high match numbers into high match quality, which makes it perform well in the case of low textures and blurs. R2D2 [18] proposes a predictor of the local descriptor discriminativeness, which can ignore ambiguous areas, thus leading to reliable keypoint detection and description. The above-mentioned local features use the nearest neighbor search to find matches between images. Recently, SuperGlue [12] proposed a network flexible context aggregation mechanism based on attention, which achieves impressive performance by learning feature matching via graph neural networks. However, these methods rely on the detector and cannot guarantee that detected feature points from images share landmarks.

### 2.2. Detector-Free Local Feature Matching

Detector-free methods do not require a detector and directly describe and match in a unified manner. NCNet [13] proposed a different approach by directly learning the dense correspondences in an end-to-end manner. It constructs 4D cost volumes to enumerate all the possible matches between the images and uses 4D convolutions to regularize the cost volume and enforce neighborhood consensus among all the matches. DGC-Net [23] and GLU-Net [24] are based on the optical flow method, which can provide a dense matching effect in the case of large transformation. DualRC-Net [14] obtains pixel-wise correspondences in a coarse-to-fine manner which extracts both coarse- and fine-resolution feature maps. A full but coarse 4D correlation tensor is produced by a coarse map, which will be refined by a learnable neighborhood consensus module. The fine-resolution feature maps are used to obtain the final dense correspondences guided by the refined coarse 4D correlation tensor. ASpanFormer [25] is built on the hierarchical attention structure, adopting a novel attention operation that is capable of adjusting attention span in a self-adaptive manner. LoFTR [15] describes the image by a transformer that first establishes

pixel-wise dense matches at a coarse level and later refines the matches. COTR [16] first downsamples each image with a CNN, and then the result is fed into a transformer along with the query point. The work of this paper is inspired by LoFTR [15] and COTR [16] in terms of using self- and cross-attention for message passing, but a fine-level regression is proposed to obtain more accurate sub-pixel coordinates, and it can be guaranteed that the corresponding coordinates will share the 3D points.

### 2.3. Transformers in Vision

Due to its parallelism, high computational efficiency, and more interpretable model, Transformer has become the most popular model in the field of natural language processing (NLP). Recently, in order to apply Transformers to vision tasks, a lot of related studies have appeared, such as Transformers in image segmentation [26–29], Transformers in object detection [30–34], Transformers in image classification [35–37], and Transformers in image enhancement [38,39]. Concurrently with our work, some researchers [15,16,40] also apply Transformers to the field of local feature matching. The Transformer can improve the accuracy of matching by capturing the context information between two images well.

## 3. Method

### 3.1. Problem Definition

The image-matching problem is formulated as determining a set of point coordinates $\mathbf{C}^t$ in the target image $I^t$ that matches guided points $\mathbf{C}^s$ in the source image $I^s$, as described in Equation (1).

$$\mathbf{C}^t = \mathcal{M}_{s \to t}(\mathbf{C}^s | I^s, I^t) \tag{1}$$

While a straightforward solution to this problem is by directly regressing $\mathcal{M}_{s \to t}$ [16], the problem is proposed to be efficiently solved with more accurate regression in two stages. At the first stage, images are subdivided into coarse-level patches $\mathcal{P}^s$ and $\mathcal{P}^t$ at $N \times N$ resolution for both $I^s$ and $I^t$. The aim is to learn an embedding for each patch $\mathbf{p}$ so that the angles between latent feature vectors of patches measure their similarity. Specifically, the patch similarity between two patches is defined as:

$$S(\mathbf{p}_a, \mathbf{p}_b) = \mathcal{F}(\mathbf{p}_a) \cdot \mathcal{F}(\mathbf{p}_b). \tag{2}$$

$\mathcal{F}$ is the patch embedding that maps a patch to a unit feature vector in the latent space.

When the patch match is established based on the similarity score, the coordinate regression problem is solved in a manner similar to Equation (1) but in a local manner:

$$\mathbf{c}_i^t = \mathcal{M}(\mathbf{c}_i^s | \mathbf{p}_i^s, \mathbf{p}_i^t) \quad \forall \mathbf{c}_i^s \in \mathbf{C}^s. \tag{3}$$

Specifically, $\mathcal{M}$ is learned as a coordinate regression function that maps each guided point $\mathbf{c}_i^s$ in $\mathbf{C}^s$ to a target point coordinate $\mathbf{c}_i^t$ in $\mathbf{C}^t$. $\mathbf{p}_i^s \in \mathcal{P}^s$ is the patch in the source image where $\mathbf{c}_i^t$ locates in. $\mathbf{p}_i^t \in \mathcal{P}^t$ is the most similar patch to $\mathbf{p}_i^s$ and can be obtained from Equation (2).

Figure 2 illustrates GLFNet architecture as the solution to the problem. The feature extraction network (Figure 2a) is introduced in Section 3.2. It serves as a basis for the coarse-level matching network (Figure 2b) for Equation (2), which will be described in Section 3.3. Section 3.4 shows the fine-level regression network (Figure 2c) that implements Equation (3). More implementation details are finally included in Section 3.5.

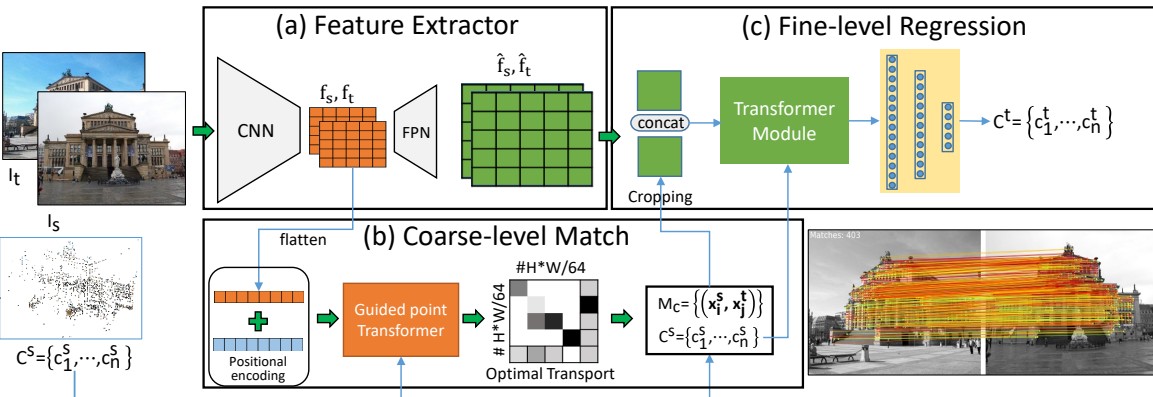

**Figure 2. The GLFNet architecture.** The proposed method is made of three components: (**a**) the feature extractor, which can extract the input features of coarse-level match network and fine-level regression; (**b**) the coarse-level match network, which can make the images divided into multiple patches to calculate the coarse match prediction; and (**c**) the fine-level regression, which can obtain the fine regression coordinates.

### 3.2. Feature Extraction Network

First, the source and target images $I^s$ and $I^t$ are passed through a feature extraction network in Figure 2a to derive image features in both low resolution and high resolution. Specifically, each image is downsampled through a CNN [41] to obtain a coarse feature map, where each feature $\mathbf{f_p}$ is corresponding to a subdivided patch $\mathbf{p}$ in $\mathcal{P}^s$ or $\mathcal{P}^t$. Then, the coarse feature map is upsampled back to one half of the original resolution using an FPN network [42] to derive the fine feature $\hat{\mathbf{f}}_\mathbf{c}$ for each pixel $\mathbf{c}$. The feature extraction network serves as an image backbone, where the coarse features $\mathbf{f_p}$ are further processed to match coarse patches and $\hat{\mathbf{f}}_\mathbf{c}$ are analyzed during coordinate regression.

### 3.3. Coarse-Level Match Network

Figure 2b illustrates the coarse-level match network that matches patches in the target image to those in the source image. To implement the patch embedding $\mathcal{F}$ in Equation (2), a transformer architecture with self-attention and cross-attention layers is used to consider coarse features $\mathbf{f_p}$ in both source and target images.

#### 3.3.1. Positional Encoding

First, the patch coordinate is encoded with a 2D extension of the standard positional encoding [43] as follows:

$$\mathcal{PE}_\mathbf{p}^m = f(x,y)^m := \begin{cases} \sin(\omega_k \cdot x), & m = 4k \\ \cos(\omega_k \cdot x), & m = 4k+1 \\ \sin(\omega_k \cdot y), & m = 4k+2 \\ \cos(\omega_k \cdot y), & m = 4k+3 \end{cases} \tag{4}$$

$\omega_k = \frac{1}{10,000^{\frac{2k}{d}}}$, $(x,y)$ is the coordinate of a patch $\mathbf{p}$, $d$ is the number of dimensions of the encoded feature, and $\mathcal{PE}_\mathbf{p}^m$ represents the value of its $m$-th feature dimension. The coarse feature $\mathbf{f_p}$ is added with its positional encoding for patch $\mathbf{p}$, which enriches image features with spatial information. For each image, all patch features are stacked as a matrix $^{(0)}\mathbf{x}$, representing the input to the first transformer module.

### 3.3.2. Guided Point Transformer

Our transformer module in Figure 2b consists of several attention layers [43]. The *i*-th attention layer can be described as mapping a query $\mathbf{q}_i$ and a set of key–value pairs $(\mathbf{k}_j, \mathbf{v}_j)$ to an output. $\mathbf{q}_i$, $\mathbf{k}_j$, and $\mathbf{v}_j$ can be expressed as follow:

$$\mathbf{q}_i = \mathbf{W}_1^{(\ell)}\mathbf{x}_i + \mathbf{b}_1, \tag{5}$$

$$\left[\begin{array}{c} \mathbf{k}_j \\ \mathbf{v}_j \end{array}\right] = \left[\begin{array}{c} \mathbf{W}_2 \\ \mathbf{W}_3 \end{array}\right]^{(\ell)}\mathbf{x}_j + \left[\begin{array}{c} \mathbf{b}_2 \\ \mathbf{b}_3 \end{array}\right]. \tag{6}$$

$\mathbf{W}_*$ and $\mathbf{b}_*$ are parameters to learn, $\ell$ stands for the $\ell$-th attention layer, and $^{(\ell)}\mathbf{x}$ represents the output from the $(\ell-1)$-th layer and input to the $\ell$-th layer. Next, the message $\mathbf{m}_i$ is obtained by weighting and aggregation through the attention mechanism:

$$\mathbf{m}_i = \sum_{j=1}^{|\mathcal{P}|} \alpha_{ij}\mathbf{v}_j, \tag{7}$$

where the attention weight $\alpha_{ij} = \text{Softmax}(\mathbf{q}_i^\top \mathbf{k}_j)$. Finally, the feature for the next layer is outputted:

$$^{(\ell+1)}\mathbf{x}_i = {}^{(\ell)}\mathbf{x}_i + \text{MLP}\left(\left[{}^{(\ell)}\mathbf{x}_i\|\mathbf{m}_{\mathcal{E}\to i}\right]\right), \tag{8}$$

where $[\cdot \| \cdot]$ denotes concatenation.

To enhance performance, a combination of self-attention and cross-attention mechanisms is employed to capture descriptive patch features through multiple attention layers, as illustrated in Figure 3. Building upon the traditional transformer architecture, a novel approach called guided point transformer is introduced. Initially, the extracted features are passed through a feature extractor, resulting in coarse feature maps $\mathbf{f_s}$ and $\mathbf{f_t}$. These maps are then transformed into one-dimensional vectors and augmented with positional coding. Next, the self-attention and cross-attention layers are employed to exploit high-dimensional information within and between the images, producing output features $\mathbf{f_s'}$ and $\mathbf{f_t'}$. To incorporate information from guided points, these points are encoded using positional encoding, and a cross-attention layer is employed to decode the guided points' features $\mathbf{Q}s$ from the feature map $\mathbf{f_s'}$. Subsequently, another cross-attention layer is used to extract information between $\mathbf{Q}s$ and $\mathbf{f_t'}$, resulting in $\mathbf{Q}s'$ and $\mathbf{f_t''}$ for subsequent optimal transport operation.

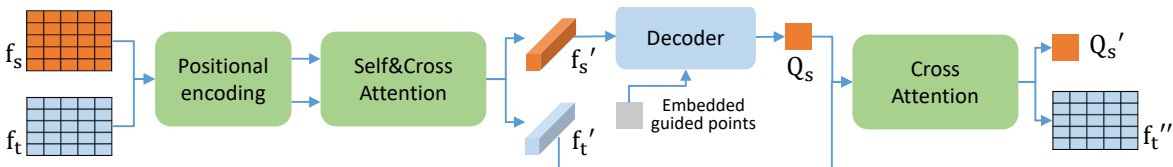

**Figure 3. The guided point transformer.** The coarse feature maps $\mathbf{f_s}$ and $\mathbf{f_t}$ are transformed into one-dimensional vectors with positional coding and sent to self-attention and cross-attention layers. Then, the embedded guided points and $\mathbf{f_s'}$ are sent to the decoder to decode the $\mathbf{Q}s$. Subsequently, another cross-attention layer is used to extract information between $\mathbf{Q}s$ and $\mathbf{f_t'}$.

### 3.3.3. Coarse-Level Matching

After the transformer module, the features for both the source guided points and target images are obtained as $\{\mathbf{x}_i^s\}$ and $\{\mathbf{x}_j^t\}$. The matches between source and target are established by solving an optimal transport problem following SuperGlue [12] using Sinkhorn algorithm [44]. The pairwise similarity is computed scores as:

$$\mathbf{S}_{i,j} = <\mathbf{x}_i^s, \mathbf{x}_j^t>, \forall(i,j) \in |\mathbf{C}^s| \times |\mathcal{P}^t|, \tag{9}$$

where $< \cdot, \cdot >$ is the inner product. Then, the optimal transport problem is solved to obtain an assignment matrix $\mathbf{A}_{i,j}$ by maximizing the total score $\sum_{i,j} \mathbf{S}_{i,j} \mathbf{A}_{i,j}$. Then, the mutual nearest neighbor (MNN) is used to find the nearest neighbor of the assignment matrix $\mathbf{A}$. The coarse-level matches can be obtained:

$$\mathcal{M}_c = \left\{ (\mathbf{x}_i^s, \mathbf{x}_j^t) \mid \forall (i, j) \in \mathrm{MNN}(\mathbf{A}), \mathbf{A}(i, j) \geq \theta_c \right\}, \tag{10}$$

where $\theta_c$ is the threshold based on which low-quality matches are filtered out. To handle patches without valid matches, a dustbin is added for each image [12] so that the optimal transport establishes pseudo-matches between these patches and dustbins.

### 3.4. Fine-Level Regression

The guided points $\mathbf{c}_i^s \in \mathbf{C}^s$ are divided by downsampling factors to obtain their respective coordinates on the coarse and fine feature maps. By utilizing the guided points' coordinates on the coarse feature map, the patches can be determined to which they belong. Subsequently, the corresponding patches of each guided point on the fine feature map can be determined based on coarse-level matches $\mathcal{M}_c$. To enhance the accuracy of subsequent window ranges, the center coordinates of each patch are adopted instead of the top-left coordinates. Precisely, the coordinates of the corresponding patch are adjusted by adding 0.5 and multiplying by the downsampling factor, yielding the coarse correspondence coordinates $\bar{\mathbf{c}}_i^t$ on the target image. In the fine-level regression stage, the accurate coordinate $\mathbf{c}_i^t$ is regressed given context information of neighborhood at $\mathbf{c}_i^s$ in the source and $\bar{\mathbf{c}}_i^t$ in the target feature maps. For each location $\mathbf{c}$ at the original resolution, its neighborhood is represented as a $9 \times 9$ window centered at $\mathbf{c}$. For each location in the window, the bilinear sampling is performed at both the coarse feature map $\mathbf{f_p}$ and the fine feature map $\hat{\mathbf{f}}_\mathbf{c}$ from the feature extractor. All collected features from $9 \times 9$ locations are aggregated as the point feature for $\mathbf{c}$. The point features for $\mathbf{c}_i^s$ and its coarse correspondence $\bar{\mathbf{c}}_i^t$ are further concatenated to represent the context information for the $i$-th input point.

A transformer module is utilized, which contains a self-attention layer and a cross-attention layer, to further enrich the context information. In detail, the query is provided as the positional encoding of the input point coordinates $\mathbf{c}_i^s$. The context information for the input point from two images is stacked as a matrix and pass them as the key–value pair to the network. The output of the transformer module is finally passed through a 3-layer MLP to predict the offsets of the target points to $\bar{\mathbf{c}}_i^t$. The offsets are added to $\bar{\mathbf{c}}_i^t$ to obtain the final correspondence coordinates $\mathbf{c}_i^t$. As shown in Figure 4, a coarse-to-fine approach is introduced, which effectively combines the advantages of both methods, ensuring both efficiency and performance benefits.

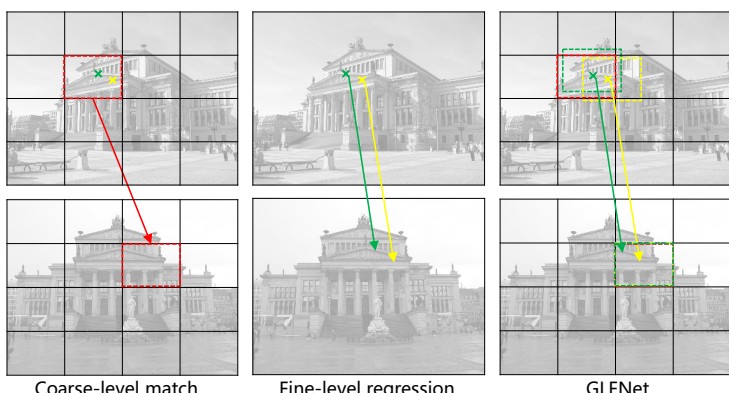

Coarse-level match      Fine-level regression      GLFNet

**Figure 4.** Coarse matching alone will lead to fast matching speeds but large errors. Direct global regression such as COTR [16] is very slow and inaccurate due to the large range of regression. GLFNet narrows the search space based on the results of coarse matching and performs regression in a small range, taking into account both accuracy and speed.

*3.5. The Implementation Details*

In order to train the network, two losses are designed to guide the coarse-level match network and the fine-level regression network. First, the primary objective of the network is to generate an assignment matrix that aligns with the ground truth. Therefore, the assignment matrix $\mathbf{A}_{i,j}$ is penalized for matches between patches and dustbins.

$$
\begin{aligned}
\mathcal{L}_c = - \sum_{(i,j)\in\mathcal{M}_c^{\text{gt}}} \log \mathbf{A}_{i,j} \\
- \sum_{i\in\mathcal{I}} \log \mathbf{A}_{i,d} - \sum_{j\in\mathcal{J}} \log \mathbf{A}_{d,j}
\end{aligned}
\tag{11}
$$

$d$ represents the dustbin patch, $\mathcal{M}_c^{\text{gt}}$ is the correspondence matches from the ground truth, and $\mathcal{I}$ and $\mathcal{J}$ are patches without matches from source and target images in the ground truth.

The regression network is additionally supervised given ground-truth corresponding points $\mathbf{c}_i^{t,\text{gt}}$ using mean squared errors.

$$
\mathcal{L}_f = \frac{1}{|\{\mathbf{c}_i^t\}|} \sum ||\mathbf{c}_i^t - \mathbf{c}_i^{t,\text{gt}}||^2,
\tag{12}
$$

The final loss includes the losses for both terms: $\mathcal{L} = \mathcal{L}_c + \mathcal{L}_f$. The GLFNet is trained on the MegaDepth dataset [45], which provides both images and corresponding dense depth maps, generated by SfM [46]. The proposed model is trained using Adam optimizer with an initial learning rate of $1 \times 10^{-3}$ and a batch size of 8. The model converges after 72 h of training on 8 GTX 3090 GPUs. A modified version of ResNet-18 [41] is used as the feature extractor. The entire model is trained end-to-end with randomly initialized weights. The number of coarse match layers is set to 4, and the number of adaptive fine regression layers is set to 1. The threshold $\theta_c$ of the assignment matrix is set to 0.2.

## 4. Experiments

Initially, the effectiveness of the two primary components and the parameter sensitivity of the proposed GLFNet is evaluated in Section 4.1. Subsequently, experimental evaluations are conducted to determine the contribution of guided points in Section 4.2. Finally, a comparison of the proposed method with state-of-the-art techniques across multiple tasks is presented. Specifically, Section 4.3.1 examines the performance of remote-sensing image registration, while Section 4.3.2 compares reconstruction registration performance. Additionally, Section 4.3.3 evaluates the performance of motion image analysis in the context of optical flow estimation, and Section 4.3.4 assesses the performance of visual localization.

*4.1. Ablation Study*

In order to evaluate the effectiveness of the two main components (coarse-level match network and fine-level regression) in the proposed GLFNet, an ablation study is conducted on HPatches [47]. Furthermore, to gauge the sensitivity of GLFNet to its parameters, a sensitivity analysis is conducted on HPatches [47] focusing on the main parameters.

4.1.1. Ablation Study

Table 1 shows the comparison of different components of GLFNet. Since the coarse-level match network only matches each patch on the downsampled image with the guided point, leading to a relatively lower accuracy when solely using this module. Furthermore, the performance of the transformer module in the coarse-level match network is compared, demonstrating the clear superiority of the guided point transformer over the conventional transformer.

**Table 1.** Ablation study on HPatches [47].

| Method | Homography Est.AUC | | | Time Cost |
|---|---|---|---|---|
| | **@3 px** | **@5 px** | **@10 px** | |
| Coarse-level match network (conventional transformer [15]) | 14.9 | 31.7 | 56.4 | 1 min 3 s |
| Coarse-level match network (guided point transformer) | 18.7 | 36.0 | 61.2 | 52 s |
| Fine-level regression | 54.0 | 68.2 | 77.4 | 1 h 12 min |
| GLFNet | 68.2 | 77.9 | 86.5 | 1 min 24 s |

Fine-level regression regresses coordinates on a sub-pixel scale, so it is competitive to take only this module. However, based on the coarse-level match network, GLFNet performs fine-level regression, which integrates the advantages of the two modules and achieves high accuracy. For the time cost, as shown in Table 1, the coarse-level match network is fast. Due to the large range of regression, the speed of fine-level regression is slow. However, GLFNet narrows the search space by the coarse-level match network, which can ensure high speed while maintaining high accuracy.

4.1.2. Parameter Sensitivity Analysis

To investigate the impact of various parameters on the proposed method, the parameter sensitivity experiments are conducted on key parameters and the results are presented in Table 2. In terms of the window size for fine-level regression, it is discovered that $9 \times 9$ yields optimal accuracy while also considering computational efficiency. When reducing the size from $9 \times 9$, a significant drop in GLFNet's matching accuracy is observed, albeit with a slight improvement in efficiency. Conversely, increasing the size beyond $11 \times 11$ leads to a notable reduction in both the efficiency and accuracy of GLFNet. The reason behind this lies in the fact that smaller sizes result in reduced regression areas, thus decreasing the algorithm's running time, but at the expense of decreased precision in returning accurate coordinates. Conversely, larger sizes expand the regression area and increase the algorithm's running time without proportionally enhancing regression accuracy. As for the parameter $\theta_c$ used in coarse-level matching, it is determined that 0.2 serves as the optimal threshold. Altering $\theta_c$ in either direction results in reduced model accuracy, emphasizing the significance of the chosen value. Interestingly, different $\theta_c$ values exhibit no significant difference in GLFNet's operational efficiency due to the swift operation of the matching filter module.

**Table 2.** Parameter sensitivity experiment on HPatches [47].

| Parameter | Homography Est.AUC | | | Time Cost |
|---|---|---|---|---|
| **Window Size of Fine-Level Regression** | **@3 px** | **@5 px** | **@10 px** | |
| $5 \times 5$ | 67.3 | 76.9 | 84.7 | 1 min 3 s |
| $7 \times 7$ | 67.9 | 77.5 | 86.3 | 1 min 11 s |
| $9 \times 9$ | 68.2 | 77.9 | 86.5 | 1 min 24 s |
| $11 \times 11$ | 68.0 | 78.1 | 85.6 | 2 min 06 s |
| $\theta_c$ of coarse-level matching | @3px | @5px | @10px | |
| 0.1 | 67.6 | 77.2 | 85.9 | 1 min 26 s |
| 0.2 | 68.2 | 77.9 | 86.5 | 1 min 24 s |
| 0.3 | 67.7 | 77.1 | 85.7 | 1 min 24 s |
| 0.4 | 67.4 | 76.8 | 85.5 | 1 min 22 s |
| 0.5 | 67.0 | 76.2 | 84.9 | 1 min 21 s |

*4.2. Experimental Evaluations*

4.2.1. The Contribution of Guided Points

The problem is reformulated as corresponding feature extraction in the target image given guided points from the source image as explicit guidance. So, the contribution of guided points as input is intended to be demonstrated. As shown in Figure 5, this is a scene with big differences in perspective. The SuperPoint [10] is adopted to extract points on the

source image as guided points. SuperPoint+SuperGlue [10,12] extracts feature points on two images, respectively, and then performs matching between point pairs. COTR [16] performs a global search on the target image for each guided point. GLFNet first matches the patches of the two images based on the guided point transformer and then regresses the exact coordinates based on guided points. Therefore, the qualitative example proves that guided points are contributive, and GLFNet can use guided points efficiently.

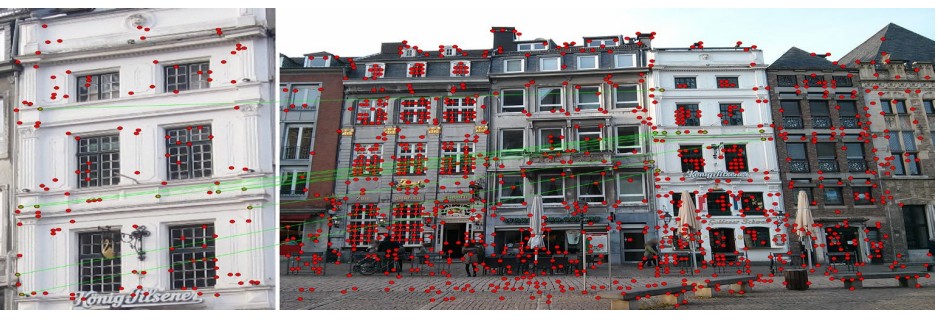

a. SP+SuperGlue

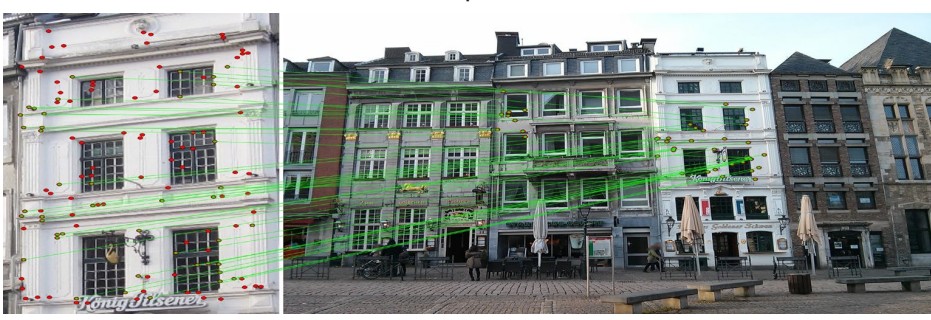

b. COTR

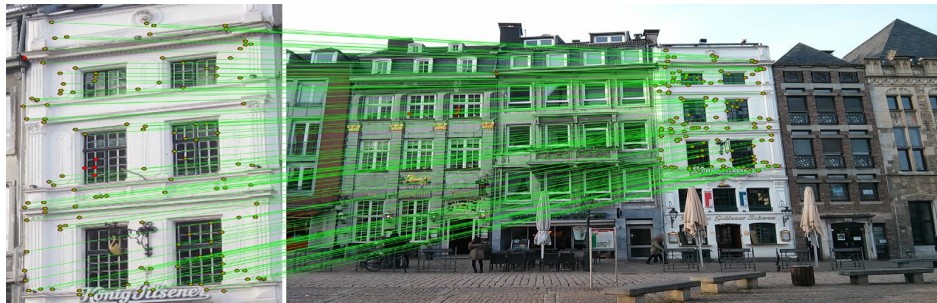

c. GLFNet

**Figure 5. Comparison between the proposed GLFNet, COTR [16], and SuperPoint + Super-Glue [10,12] in the case of scenes with large differences in perspective.** The results clearly demonstrate that GLFNet effectively leverages guided points and performs admirably in scenarios characterized by large differences in perspective.

4.2.2. Image Matching with Guided Points

Since the contribution of the guiding points has been demonstrated, the image matching is completed under this premise. HPatches [47] is a very popular dataset to evaluate image matching, which contains 57 scenes under photometric changes and 59 sequences that show significant geometric deformations due to viewpoint change. Because the problem is reformulated, methods that can solve this problem are selected for comparison. The baseline includes two categories of methods: (1) detector-based image-matching methods: D2Net + NN [9], R2D2 + NN [18], and SuperGlue + SuperPoint [10,12]; (2) a detector-free image-matching method: COTR [16]. To ensure a fair comparison between methods that produce different numbers of matches, the corner error between images is computed. Meanwhile, three different guided points are adopted for comparison to

prove the superiority of the proposed method. LoFTR and ASpanFormer are also included in the comparison, which are the SOTA detector-free methods but cannot solve the reformulated problem.

As shown in Table 3 and Figure 6, GLFNet has a significant improvement in matching accuracy under all error threshold conditions. Specifically, when the error threshold is smaller, the improvement of GLFNet is more significant. At the same time, GLFNet is better than most methods in time cost while maintaining high accuracy. In particular, compared with the COTR method, which can also take guided points as input, the time cost has been improved exponentially. By the way, GLFNet also outperforms LoFTR and ASpanFormer in time cost and accuracy.

**Table 3. Image matching on HPatches [47].** The AUC of the corner error in percentage is reported. The time measurement experiments for all algorithms were run on the same machine.

| Category | Method | Homography Est.AUC | | | Time Cost | #Matches |
|---|---|---|---|---|---|---|
| | | @3 px | @5 px | @10 px | | |
| Detector-based | SIFT + NN [8] | 49.8 | 61.0 | 73.9 | 47 s | 0.4K |
| | D2Net + NN [9] | 23.2 | 35.9 | 53.6 | 8 min 57 s | 0.2 K |
| | R2D2 + NN [18] | 50.6 | 63.9 | 76.8 | 4 min 22 s | 0.5 K |
| | SuperPoint + SuperGlue [12] | 53.9 | 68.3 | 81.7 | 1 min 11 s | 0.6 K |
| Detector-free | LoFTR [15] | 65.9 | 75.6 | 84.6 | 5 min 2 s | 1.0 K |
| | ASpanFormer [25] | 67.4 | 76.9 | 85.6 | 6 min 1 s | 1.0 K |
| | SIFT + COTR [16] | 34.5 | 49.8 | 67.2 | 4 h 1 min | 1.0 K |
| | R2D2 + COTR [16] | 34.9 | 50.2 | 68.7 | 4 h 3 min | 1.0 K |
| | SuperPoint + COTR [16] | 56.1 | 69.4 | 81.3 | 4 h 2 min | 1.0 K |
| | SIFT + GLFNet | 62.4 | 73.1 | 83.7 | 1 min 10 s | 1.0 K |
| | R2D2 + GLFNet | 63.8 | 74.7 | 84.2 | 1 min 30 s | 1.0K |
| | SuperPoint + GLFNet | 68.2 | 77.9 | 86.5 | 1 min 24 s | 1.0 K |

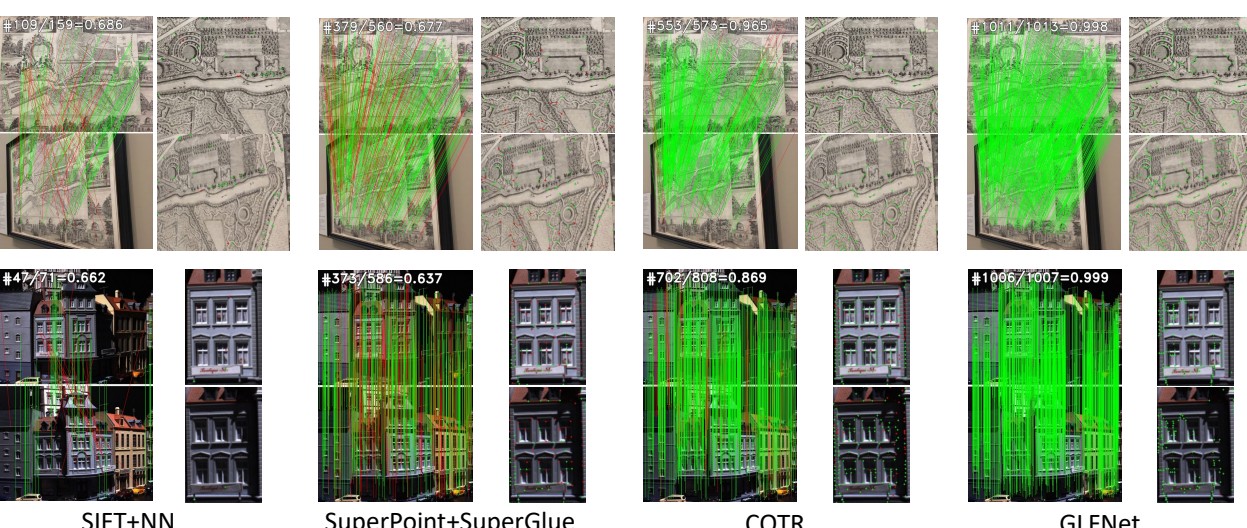

SIFT+NN    SuperPoint+SuperGlue    COTR    GLFNet

**Figure 6. Qualitative image-matching results of Hpatches [47].** The image pair on the left displays the image-matching results, with the correct matching rate (CMR) calculation shown in the upper left. The image pairs on the right exhibit enlarged sub-results, where green feature points denote correct matching, and red feature points signify incorrect matching. Notably, GLFNet demonstrates superior performance compared to all the comparison methods.

*4.3. Comparison*

4.3.1. Remote-Sensing Image Registration

Remote-sensing image registration represents a critical and fundamental step in the process of aligning images of the same scene, captured by either the same or different

sensors, at different times, and/or from varying perspectives. The primary objective of remote-sensing image registration is to achieve accurate alignment between the target and source images. This field finds broad applications, including change detection, landscape planning, and agriculture monitoring. The image registration pipeline typically encompasses feature extraction, feature matching, transformation model estimation, and image transformation. Among these stages, feature matching is particularly vital, making GLFNet an effective approach for enhancing this task.

Datasets

The Google Earth dataset [1] is employed to assess the efficacy of GLFNet in remote-sensing image registration. Comprising over 9000 high-resolution satellite images acquired across multiple seasons, the dataset primarily covers the Greater Boston area. The input and template images exhibit notable differences in traffic conditions and vegetation, posing a significant challenge to feature matching.

Comparison

The performance of the proposed method is evaluated by comparing it with two classical methods, SIFT + RANSAC [8] and CLKN [48], and three state-of-the-art methods, DHN [49], MHN [50], and DLKFM [1]. Vanilla is also included as a baseline, which predicts the centers of four corner boxes without any prior knowledge. The evaluation metric used is the average corner error in pixels, which is calculated as the L2 distance between the warped and original corners, following the methodology outlined in [1].

Results

In Table 4, Figures 7 and 8, the comparison results between GLFNet and benchmark methods are presented. The proposed method outperforms state-of-the-art methods across different error scales. Specifically, the proposed method achieves the three-pixel error threshold for most images, while the baseline methods do not. Moreover, the greater the error requirement, the more significant the improvement over the baseline methods.

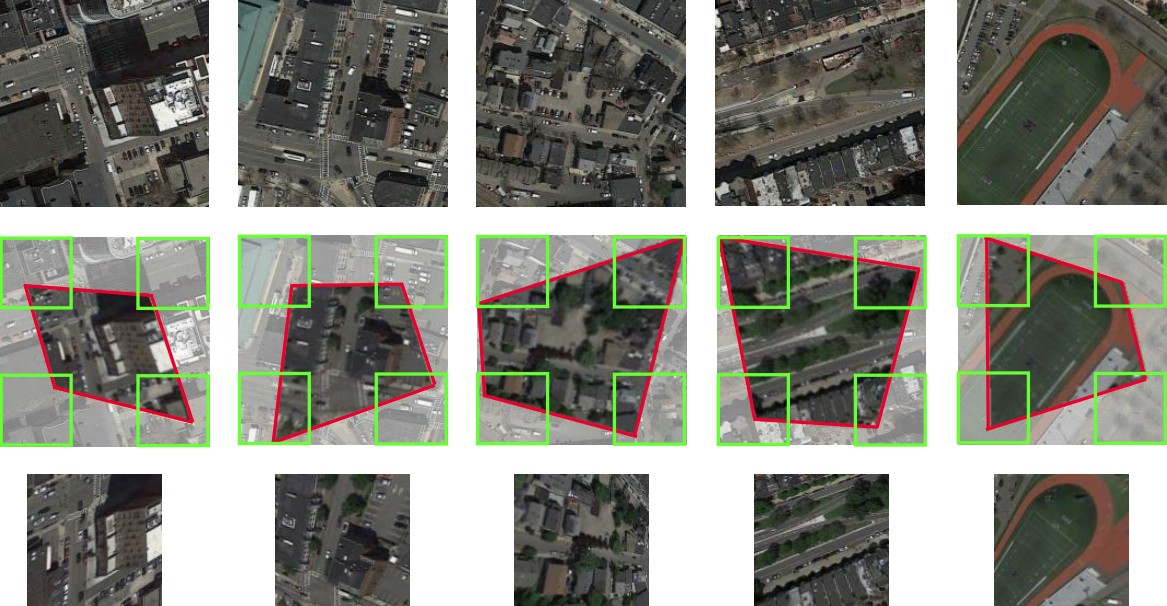

**Figure 7. Qualitative remote-sensing image registration results of Google Earth.** The topmost row displays the input image, while the bottommost row depicts the template image. GLFNet is employed to carry out matching between the input image and the template, leading to the computation of the rotation matrix. The middle row represents the region within the input image where the template image corresponds.

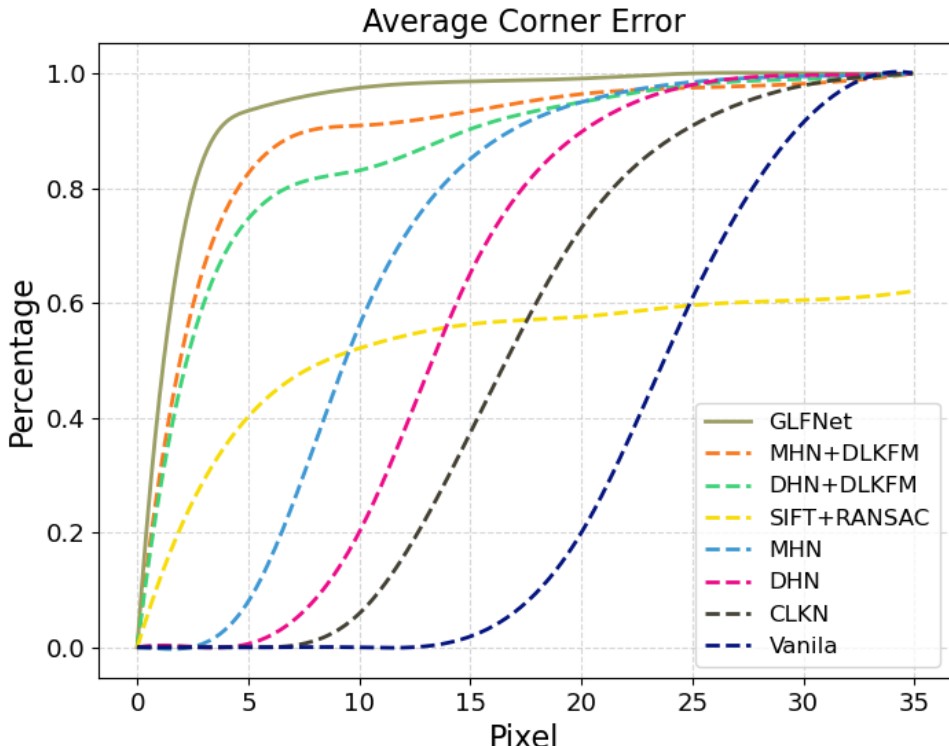

**Figure 8. Comparison of average corner error on Google Earth.** The x-axis represents the average pixel error, and the y-axis represents the proportion of images with an average pixel error lower than x. The proposed method outperforms state-of-the-art methods across different error scales.

**Table 4.** Remote-sensing image registration on Google Earth.

| Method | Average Corner Error | | |
|---|---|---|---|
| | <3 Pixel | <5 Pixel | <10 Pixel |
| Vanila | 0 | 0 | 0 |
| CLKN [48] | 0 | 0 | 5.9 |
| DHN [49] | 0 | 1 | 20.1 |
| MHN [50] | 1.0 | 8.0 | 56.1 |
| SIFT + RANSAC [8] | 29.1 | 40.2 | 52.1 |
| DHN + DLKFM [1] | 60.0 | 74.7 | 83.1 |
| MHN + DLKFM [1] | 66.3 | 82.6 | 90.9 |
| GLFNet | 85.3 | 93.5 | 97.5 |

### 4.3.2. Reconstruction Registration

With the development of 3D reconstruction, an increasing number of 3D models are being reconstructed, giving rise to the task of reconstruction registration. This registration process involves either incremental reconstruction based on a large-scale 3D model or the combination of multiple 3D models with small overlapping regions. Following the pipeline proposed in [51], the reconstruction registration procedure consists of three main steps: image retrieval, image matching, and transformation solving. In this work, GLFNet is taken as a replacement for the conventional image-matching module. The key advantage of GLFNet lies in its ability to directly utilize the existing 3D points from the 3D model as guided points, thus avoiding errors that could arise from re-extracting feature points. This enables us to obtain more accurate 3D point-matching pairs, consequently enhancing the effectiveness of subsequent transformation solutions.

Datasets

Following [51], the proposed method is evaluated on the public realistic synthetic scenarios datasets published in 1DSFM [52], which consists of twelve medium-scale datasets, a large-scale dataset Piccadilly, and a challenging dataset Gendarmenmarkt with symmetric architectures. The dataset is first divided into subsets for partial reconstruction through off-the-shelf community detection algorithms, and then partial reconstruction results for each subset are obtained using COLMAP [46] as input to the model.

Comparison

The proposed method is compared with four state-of-the-art global structure from motion (SfM) methods, two incremental SfM methods, and Merge-SfM [51]. Most methods gather all images together to re-perform SfM reconstruction, while the proposed method and Merge-SfM [51] are different from this. The standard SfM system [51] is replaced with GLFNet, and more implemented materials are provided in the supplementary material. Following [52], the calculated $N_c$, $\tilde{x}$ and $\bar{x}$ are compared with [53] to evaluate the effectiveness of registration. The units are approximately in meters, as geotags associated with images in the collection are used to place each sequential SfM reconstruction in an approximate world coordinate frame, and a RANSAC approach [54] is used to compute the absolute orientation between a candidate reconstruction and the sequential SfM solution.

Results

Table 5 shows the comparison of reconstruction results between GLFNet and other methods. GLFNet preserves all the recovered images from partial reconstructions and outperforms state-of-the-art methods in reconstruction accuracy. Specifically, the median and mean position errors of the proposed method are competitive on most datasets, which proves the success of our registration. Based on the success, the number of recovered cameras of the proposed method is significantly higher than other methods, and the most important reason is that GLFNet increases the probability of matching. A qualitative example of GLFNet is shown in Figure 9.

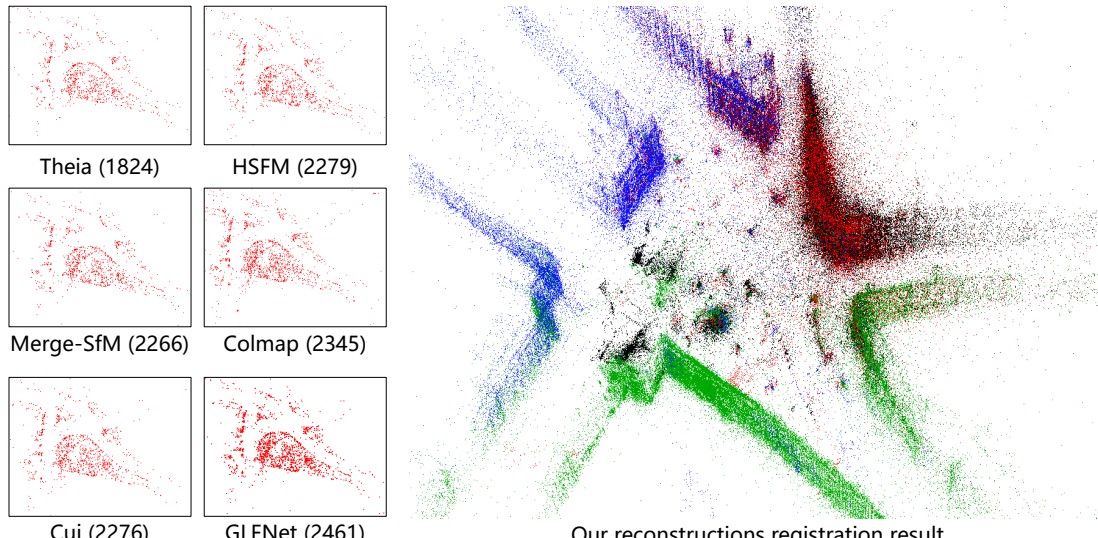

**Figure 9. Qualitative reconstructions registration result of Piccadilly.** Piccadilly consists of six partial reconstructions (each color represents a different partial 3D reconstruction), one of them is chosen as the source reconstruction, and GLFNet is used to match the guided points in the source reconstruction to the other five reconstructions. The numbers in parentheses represent the number of recovered cameras.

**Table 5. Reconstruction registration accuracy comparison on public realistic synthetic scenarios datasets** [52]. $\tilde{x}$ and $\bar{x}$ denote the median and mean position errors in meters respectively by taking the result of [53] as a reference; $N_i$ represents the cameras in the largest connected component of the input EG graph, which is published in [55]; and $N_c$ represents the recovered cameras.

| Dataset | | | 1DSfM [52] | | | LUD [55] | | | Cui [56] | | | Swe [57] | | | HSfM [58] | | | Theia [59] | | | Merge-SfM [51] | | | GLFNet | | |
|---|---|---|---|---|---|---|---|---|---|---|---|---|---|---|---|---|---|---|---|---|---|---|---|---|---|---|
| Name | N | $N_i$ | $N_c$ | $\tilde{x}$ | $\bar{x}$ | $N_c$ | $\tilde{x}$ | $\bar{x}$ | $N_c$ | $\tilde{x}$ | $\bar{x}$ | $N_c$ | $\tilde{x}$ | $\bar{x}$ | $N_c$ | $\tilde{x}$ | $\bar{x}$ | $N_c$ | $\tilde{x}$ | $\bar{x}$ | $N_c$ | $\tilde{x}$ | $\bar{x}$ | $N_c$ | $\tilde{x}$ | $\bar{x}$ |
| Alamo | 4 | 627 | 529 | 0.3 | $2 \times 10^7$ | 547 | 0.3 | 2 | 574 | 0.5 | 3.1 | 533 | 0.4 | - | 566 | 0.3 | 1.5 | 520 | 0.4 | 1.8 | 582 | 0.5 | 2.6 | 599 | 0.3 | 1.9 |
| EllisIsland | 3 | 247 | 214 | 0.3 | 3 | - | - | - | 233 | 0.7 | 4.2 | 203 | 0.5 | - | 233 | 2 | 4.8 | 210 | 1.7 | 2.8 | 232 | 0.8 | 4.4 | 238 | 0.8 | 3.3 |
| Metropolis | 2 | 394 | 291 | 0.5 | 70 | 288 | 1.5 | 4 | 317 | 3.1 | 16.6 | 272 | 0.4 | - | 344 | 1 | 3.4 | 301 | 1 | 2.1 | 328 | 1.6 | 5.3 | 335 | 1.5 | 5.3 |
| MontrealND | 2 | 474 | 427 | 0.4 | 1 | 435 | 0.4 | 1 | 452 | 0.3 | 1.1 | 416 | 0.3 | - | 461 | 0.3 | 0.6 | 422 | 0.4 | 0.6 | 374 | 0.3 | 0.8 | 453 | 0.2 | 0.8 |
| NYCLibrary | 3 | 376 | 295 | 0.4 | 1 | 320 | 1.4 | 1 | 338 | 0.3 | 1.6 | 294 | 0.4 | - | 344 | 0.3 | 1.5 | 291 | 0.4 | 1 | 336 | 0.3 | 1.3 | 351 | 0.3 | 3 |
| PiazzadelPopolo | 3 | 354 | 308 | 2.2 | 200 | 305 | 1 | 4 | 340 | 1.6 | 2.5 | 302 | 1.8 | - | 344 | 0.8 | 2.9 | 290 | 0.8 | 1.5 | 344 | 0.5 | 1.1 | 346 | 0.5 | 1.1 |
| RomanForum | 4 | 1134 | 989 | 0.2 | 3 | - | - | - | 1077 | 2.5 | 10.1 | 966 | 0.7 | - | 1087 | 0.9 | 8.4 | 942 | 0.6 | 2.6 | 1109 | 0.8 | 6.4 | 1093 | 0.4 | 1.7 |
| TowerofLondon | 4 | 508 | 414 | 1 | 40 | 425 | 3.3 | 10 | 465 | 1 | 12.5 | 409 | 0.9 | - | 481 | 0.7 | 6.4 | 439 | 1 | 1.9 | 469 | 0.7 | 5.4 | 481 | 0.6 | 5.5 |
| UnionSquare | 4 | 930 | 710 | 3.4 | 90 | - | - | - | 570 | 3.2 | 11.7 | 701 | 2.1 | - | 827 | 2.8 | 3.4 | 626 | 1.9 | 3.7 | 724 | 2.3 | 5.5 | 927 | 1.4 | 5.8 |
| ViennaCathedral | 4 | 918 | 770 | 0.4 | 2e4 | 750 | 4.4 | 10 | 842 | 1.7 | 4.9 | 771 | 0.6 | - | 849 | 1.4 | 3.3 | 738 | 1.8 | 3.6 | 823 | 0.7 | 3.5 | 906 | 0.7 | 2.6 |
| Yorkminster | 2 | 458 | 401 | 0.1 | 500 | 404 | 1.3 | 4 | 417 | 0.6 | 14.2 | 409 | 0.3 | - | 421 | 1.2 | 1.7 | 370 | 1.2 | 1.8 | 431 | 1.2 | 4.2 | 438 | 0.4 | 3.8 |
| Gendarmenmarkt | 4 | 742 | - | - | - | - | - | - | 609 | 4.2 | 27.3 | - | - | - | 611 | 2.8 | 26.3 | 597 | 2.9 | 28 | 704 | 2.4 | 38 | 729 | 2.4 | 19.3 |
| Piccadilly | 6 | 2508 | 1956 | 0.7 | 700 | - | - | - | 2276 | 0.4 | 2.2 | 1928 | 1 | - | 2279 | 0.7 | 2 | 1824 | 0.6 | 1.1 | 2266 | 0.7 | 9 | 2461 | 0.3 | 1.6 |

### 4.3.3. Optical Flow Estimation

Optical flow estimation is a fundamental component of motion image analysis, serving the purpose of estimating pixel motion between two frames to determine sparse feature sets or the displacement of all image pixels, subsequently calculating their motion vectors. This technique finds extensive applications in object detection and tracking, image dominant plane extraction, motion detection, robot navigation, and visual odometry. A crucial step in optical flow estimation involves identifying pixel correspondences between the two frames, a task adeptly addressed by GLFNet through image matching, thereby enhancing the overall process.

Datasets

Following [16], to evaluate the effect of the proposed method in real 3D scenes of optical flow estimation, the ETH3D dataset [60] is adopted, which is multi-view and contains indoor scenes and outdoor scenes captured from a moving hand-held camera. A set of sparse geometrically consistent image correspondences is provided [46].

Comparison

The proposed method is compared with three state-of-the-art image-matching methods applied to this task, which are DGC-Net [23], GLU-Net [24], and COTR [16]. Meanwhile, the proposed method is compared with three state-of-the-art optical flow methods applied to this task, which are LiteFlowNet [61], PWC-Net [62], and RAFT [63]. Specifically, image pairs are sampled from each sequence at different intervals to analyze different magnitudes of geometric transformations, and the provided points are taken as ground-truth correspondences. For each selected interval, there are approximately 500 image pairs in total. The average end-point error (AEPE) is defined as the Euclidean distance between estimated and ground truth flow fields. The AEPE at different intervals is employed as the evaluation metrics.

Results

Table 6 and Figure 10 show the comparison of optical flow estimation results between GLFNet and other methods. The proposed method outperforms the baseline method for different intervals, and the improvement is obvious when the interval is large. Because the large interval leads to large differences in the perspective of the images, which improves the difficulty of matching. However, the proposed method can also accurately match when the difference in perspective is large. As shown in Figure 11, the effectiveness of GLFNet on satellite images also is demonstrated.

**Table 6. Quantitative results for optical flow estimation.** The average end-point error (AEPE) at different sampling "rates" (frame intervals) is reported. The proposed method performs significantly better as the rate increases and the problem becomes more difficult.

| Method | AEPE | | | | | | |
|---|---|---|---|---|---|---|---|
| | Rate = 3 | Rate = 5 | Rate = 7 | Rate = 9 | Rate = 11 | Rate = 13 | Rate = 15 |
| LiteFlowNet [61] | 1.66 | 2.58 | 6.05 | 12.95 | 29.67 | 52.41 | 74.96 |
| PWC-Net [62] | 1.75 | 2.10 | 3.21 | 5.59 | 14.35 | 27.49 | 43.41 |
| DGC-Net [23] | 2.49 | 3.28 | 4.18 | 5.35 | 6.78 | 9.02 | 12.23 |
| GLU-Net [24] | 1.98 | 2.54 | 3.49 | 4.24 | 5.61 | 7.55 | 10.78 |
| RAFT [63] | 1.92 | 2.12 | 2.33 | 2.58 | 3.90 | 8.63 | 13.74 |
| COTR [16] | 1.66 | 1.82 | 1.97 | 2.13 | 2.27 | 2.41 | 2.61 |
| GLFNet | 1.49 | 1.61 | 1.78 | 1.90 | 2.07 | 2.34 | 2.65 |

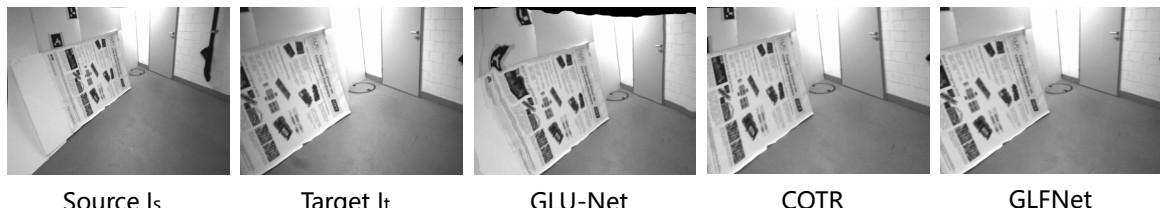

Source I_s          Target I_t          GLU-Net          COTR          GLFNet

**Figure 10. Qualitative optical flow estimation result of ETH3D.** Pairs of images from ETH3D taken by two different cameras. The proposed method significantly outperforms state-of-the-art methods.

### 4.3.4. Visual Localization

The visual localization task is to estimate the 6-DoF poses of target images with respect to the reference 3D scene model. Thus, it relies on highly robust local feature-matching methods. GLFNet is also beneficial for this task. In real-world scenarios, databases are often already described by descriptors. Traditional methods need to re-describe the base map in the database to complete visual localization, which is time-consuming. However, the proposed method can achieve a good result on this task without re-describing the base map.

### Datasets

The Aachen Day-Night dataset [64,65] is used to demonstrate the effectiveness of GLFNet for visual localization in outdoor scenes, whose base map is described by SIFT [8]. And SIFT [8] is the most common descriptor in the industry. The dataset contains 4328 source images taken during the daytime with hand-held cameras over about two years and 922 target images taken during the daytime and nighttime with mobile phone cameras. The dataset considers varying conditions, e.g., day–night changes and scene geometry changes.

### Comparison

The GLFNet is compared with state-of-the-art methods without changing the base map descriptor. And the AUC of the pose error at thresholds (0.25 m, 2°)/(0.5 m, 5°)/(5 m, 10°) is adopted as the metrics, where the pose error is defined as the maximum angular error in rotation and translation. In practice, GLFNet transfers sparse points directly from the original COLMAP [46] project into target images. To recover the camera pose, the PnP is solved from predicted matches with RANSAC [54]. The process is kept consistent for all methods.

### Results

The evaluation results of GLFNet and state-of-the-art methods are provided in Table 7. The proposed method has a significant improvement over the state-of-the-art methods in both daytime and nighttime scenarios. In the daytime scenario, the baseline methods all

work well. However, in the nighttime scenario, the gap between the proposed method and the baseline is large because the lighting difference between images is too big.

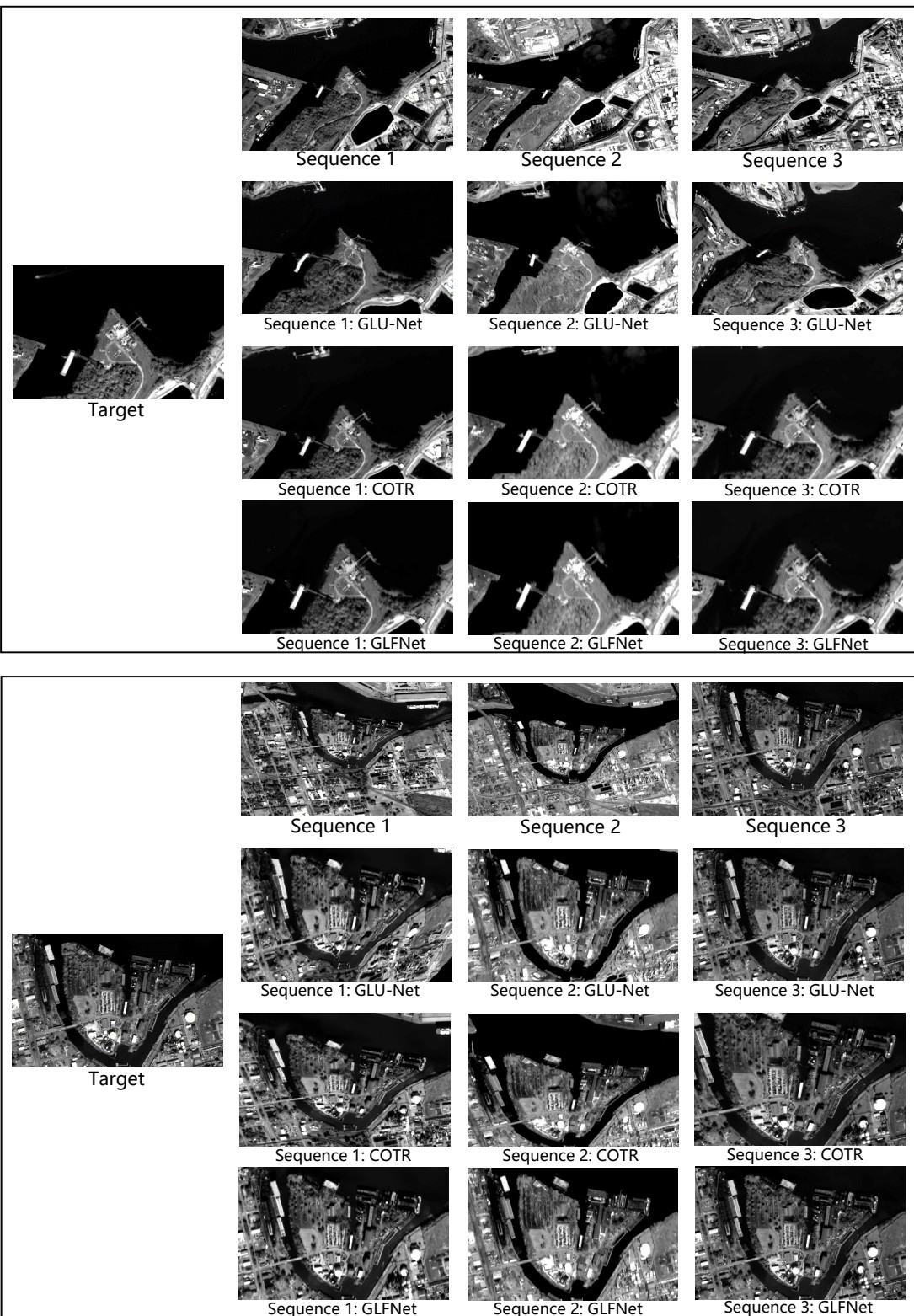

**Figure 11. Qualitative optical flow estimation result of satellite images.** GLFNet is applied to remote-sensing images, and superior results were achieved.

**Table 7.** **Visual localization evaluation on the Aachen Day-Night benchmark [64,65].** The proposed method outperforms the state-of-the-art methods.

| Matching Method | Day | Night |
|---|---|---|
| | (0.25 m, 2°)/(0.5 m, 5°)/(5 m, 10°) | |
| SIFT + NN [8] | 84.5/92.7/97.5 | 66.3/75.5/87.8 |
| SIFT + COTR [8,16] | 82.4/91.9/96.8 | 75.5/90.8/99.0 |
| SIFT + SuperGlue [8,12] | 85.3/93.9/98.2 | 72.4/88.8/96.9 |
| SIFT + GLFNet | 88.2/95.4/98.2 | 84.7/92.9/99.0 |

## 5. Discussion

As detailed in Section 4.1, the ablation experiments are conducted to assess the performance of the proposed guided point transformer in image-matching tasks in comparison to the traditional transformer. The results demonstrate the significant advantages of the guided point transformer. We observed that using the coarse-level yields faster matching speeds, although the precision is not as high. Conversely, relying solely on fine-level regression improves accuracy but sacrifices matching speed. To strike a balance between speed and accuracy, the proposed coarse-to-fine approach is introduced, which effectively combines the advantages of both methods, ensuring both efficiency and performance benefits.

As detailed in Section 4.2, the experimental evaluation of GLFNet's performance is presented. Qualitative assessments in scenarios with large perspective differences demonstrate the valuable contribution of guided points. Moreover, on the standard matching dataset Hpaches, GLFNet exhibits superior performance, attributable to three key factors. Firstly, the introduction of guided points into the image-matching problem provides valuable guidance, significantly enhancing both efficiency and accuracy. Secondly, the guided point transformer, a core component of GLFNet, effectively incorporates guided points into the feature expression process, enabling comprehensive exploration of high-dimensional information between the guided points and the target image. This fosters the extraction of informative representations crucial for achieving accurate matches. Thirdly, the network architecture design of GLFNet plays a pivotal role. By utilizing a coarse-level network, the search space is efficiently narrowed down, effectively reducing computation time. Concurrently, the fine-level network accurately regresses the coordinates, ensuring high precision in the matching process. Additionally, GLFNet exhibits remarkable flexibility, allowing seamless integration with any detector, thus broadening its potential applications across a wide range of scenarios.

As detailed in Section 4.3, the comparison of GLFNet's effectiveness across multiple tasks is presented. Notably, GLFNet achieves significant performance improvements in remote-sensing image registration compared to state-of-the-art methods. Moreover, in the reconstruction registration task, GLFNet exhibits excellent robustness across various scenarios. In the context of optical flow estimation, GLFNet outperforms all matching-based and optical-flow-based methods, with its advantage becoming particularly prominent when dealing with larger frame intervals. Additionally, GLFNet demonstrates its capacity to effectively leverage 3D points from existing 3D models, which proves beneficial for visual positioning tasks.

Despite its strengths, GLFNet does have certain limitations. The presence of dynamic objects in the image can introduce changes in their positions between the source and target images, posing significant challenges to the matching process. In future work, we aim to explore the incorporation of semantic information into the matching algorithm to mitigate the impact of dynamic objects on the matching performance.

## 6. Conclusions

This paper presents a novel approach to image matching by reformulating the problem with the guidance of guided points. This reformulation proves beneficial for tasks such as reconstruction registration, visual positioning, and incremental reconstruction, as it encourages corresponding points to share landmarks in 3D space. To address this

reformulated problem, GLFNet is proposed, which is a guided local feature-matching method that efficiently matches local features with the assistance of guided points. GLFNet comprises a coarse-to-fine network based on the guided point transformer, which enables effective search space narrowing at the coarse level and precise coordinate regression at the fine level. Extensive experiments, including ablation studies and parameter sensitivity analyses, validate the effectiveness of each proposed module in GLFNet. The results of the experimental evaluation demonstrate GLFNet's superiority in image-matching tasks. Additionally, GLFNet's performance is evaluated across various tasks, showcasing its effectiveness in downstream applications. In the future, we plan to investigate techniques to mitigate the impact of dynamic objects on image matching and explore ways to incorporate semantic information into the matching process. These efforts will further enhance the applicability and robustness of GLFNet in real-world scenarios.

**Author Contributions:** Conceptualization, S.D., Y.X. and J.H.; methodology, S.D., Y.X. and M.L.; software, M.L. and Y.X; validation, Y.X. and M.L.; formal analysis, J.H. and M.S.; investigation, S.D. and Y.X.; resources, M.S. and M.L.; data curation, M.L. and Y.X.; writing—original draft preparation, S.D. and Y.X.; writing—review and editing, J.H., M.L. and M.S.; visualization, M.S.; supervision, M.S.; project administration, M.L. All authors have read and agreed to the published version of the manuscript.

**Funding:** This research received no external funding.

**Data Availability Statement:** The links to the data sets used in this paper are as follows: HPatches: http://icvl.ee.ic.ac.uk/vbalnt/hpatches/; Google Earth: https://github.com/placeforyiming/CVPR21-Deep-Lucas-Kanade-Homography; 1DSFM: https://www.cs.cornell.edu/projects/1dsfm/; ETH3D: https://www.eth3d.net/datasets; Aachen Day-Night: https://www.visuallocalization.net/datasets/.

**Conflicts of Interest:** The authors declare no conflict of interest.

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
