# Peer review of "Guided Local Feature Matching with Transformer"

_remotesensing, doi:10.3390/rs15163989_

Round 1

Reviewer 1 Report

This paper presents a local feature matching method with transformer named GLFNet leveraging sparse feature points as guided points, which includes a coarse-level match network and a fine-level regression network. The performance of the presented method is demonstrated by various applications including remote sensing image registration, reconstruction registration, optical flow estimation, and visual localization. However, several critical concerns must be addressed before publication as follows.

1. There are several typos or missing explanations, such as “fq” in Page 5 and “xp” in Page 7. Moreover, several section titles are not rigorous and suitable. The authors should go over the entire manuscript to check these issues.

2. The description of fine-level regression in the current manuscript is not clear.

3. It is necessary to introduce more details on how to use GLFNet in image registration, 3D reconstruction and optical flow. In addition, the supplementary material is not provided in the manuscript.

4. It is necessary to provide a parameter sensitivity analysis.

5. The introduction of Figure 5 in text is missing.

The manuscript is well-written, except for some typos and format errors. Minor editing of English language is required.

Reviewer 3 Report

Dear Academic editor,

I wish to submit review for article entitled “Guided Local Feature Matching with Transformer for consideration to Sensors. I am wondering it is full length research or a review paper? I have some suggestions for better presentation of the manuscript as following:

1.     The abstract is ok. But it is not academic and need to be changed to passive form (participle). I mean for whole of manuscript too. 

2.     The introduction is short and needs to have some sentences about filling the knowledge gap, problem statement, and hypotheses for solving the problems, and also the main objectives of the study must be defined. It problem statement is very simple; which dataset should be adopted to fulfill proper and adequate outputs? Please provide motivation and justification as to why they are doing the review? The whole idea of writing this section is to cover two important questions—“What are the gaps present in the current literature?” and “Why is the current study important?”

3. Additionally, some of the paragraphs are short and must be mixed for example Paras 1 and 2. I think figure 1 is not suitable for introduction. Please move it to suitable place.

4.     Discussion is very short. For writing the discussion, please read the following sentences. Discussion: Authors should discuss the results and how they can be interpreted from the perspective of previous studies and of the working hypotheses. The findings and their implications should be discussed in the broadest context possible and the limitations of the work highlighted. It is also essential to discuss the strengths and limitations of one’s study. Comments on sources of uncertainty and error are appropriate for most papers. Future research directions may also be highlighted.

5. Conclusions section is very short and need to be improved academically. The present conclusions is only a summary with some points. Please read the following sentences for reconstructing the conclusions.  A conclusion is the final paragraph of a research paper and serves to help the reader understand why your review should matter to them. The conclusion of a manuscript should: 1. Restate your topic and why it is important 2. Restate your thesis/claim, 3. Address opposing viewpoints and explain why readers should align with your position, 4. Include a call for action or overview of future research possibilities

Moderate editing of English language required. Passive form (participle) is required for whole of the revised manuscript. 

Round 2

Reviewer 1 Report

The authors have addressed all my comments, and the revised manuscript is ready for publication.

Reviewer 2 Report

Accept in present form

Accept in present form

Reviewer 3 Report

Accept